

# N-mixture models provide informative crocodile (*Crocodylus moreletii*) abundance estimates in dynamic environments

José António Lemos Barão-Nóbrega[1,2], Mauricio González-Jaurégui[3] and Robert Jehle[2]

[1] Operation Wallacea, Spilsby, Lincolnshire, United Kingdom
[2] School of Science, Engineering and Environment, University of Salford, Salford, Greater Manchester, United Kingdom
[3] Universidad Autónoma de Campeche, Centro de Estudios de Desarrollo Sustentable y Aprovechamiento de la Vida Silvestre, Campeche, Campeche, Mexico

Corresponding authors
José António Lemos Barão-Nóbrega,
jose.antonio@opwall.com
Mauricio González-Jaurégui,
mauglezj@gmail.com

## ABSTRACT

Estimates of animal abundance provide essential information for population ecological studies. However, the recording of individuals in the field can be challenging, and accurate estimates require analytical techniques which account for imperfect detection. Here, we quantify local abundances and overall population size of Morelet's crocodiles (*Crocodylus moreletii*) in the region of Calakmul (Campeche, Mexico), comparing traditional approaches for crocodylians (Minimum Population Size—MPS; King's Visible Fraction Method—VFM) with binomial *N*-mixture models based on Poisson, zero-inflated Poisson (ZIP) and negative binomial (NB) distributions. A total of 191 nocturnal spotlight surveys were conducted across 40 representative locations (hydrologically highly dynamic aquatic sites locally known as aguadas) over a period of 3 years (2017–2019). Local abundance estimates revealed a median of 1 both through MPS (min–max: 0–89; first and third quartiles, $Q_1$–$Q_3$: 0–7) and VFM (0–112; $Q_1$–$Q_3$: 0–9) non-hatchling *C. moreletii* for each aguada, respectively. The ZIP based *N*-mixture approach shown overall superior confidence over Poisson and NB, and revealed a median of 6 ± 3 individuals (min = 0; max = 120 ± 18; $Q_1$ = 0; $Q_3$ = 18 ± 4) jointly with higher detectabilities in drying aguadas with low and intermediate vegetation cover. Extrapolating these inferences across all waterbodies in the study area yielded an estimated ~10,000 (7,000–11,000) *C. moreletii* present, highlighting Calakmul as an important region for this species. Because covariates enable insights into population responses to local environmental conditions, *N*-mixture models applied to spotlight count data result in particularly insightful estimates of crocodylian detection and abundance.

## INTRODUCTION

Measures of population abundance are key to understanding the ecology and natural history of wild animals, and form a main basis for the implementation of conservation

management plans. However, due to elusive behaviours and logistic constraints, researchers are often unable to record all individual animals in a given location. Because detectability also interacts with for example local environmental conditions, precise abundance estimates based on survey data alone are generally difficult to obtain (*e.g. Mazerolle et al., 2007*; *Sutherland, 2006*).

An emerging approach to estimate population sizes from repeated standard count surveys is represented by *N*-mixture models, which jointly estimate a measure of abundance ($\lambda$) with the probability of detecting an individual (*p*) (*Denes, Silveira & Beissinger, 2015*; *Kéry & Royle, 2016*; *Royle, 2004*; *Zipkin et al., 2014*). Binomial *N*-mixture models (*Kéry, Royle & Schmid, 2005*; *Royle, 2004*), for example, treat $\lambda$ as a random independent variable generated from a statistical distribution to estimate *p* (*Kéry & Royle, 2016*). *N*-mixture models are particularly promising for wildlife studies because they have the potential to produce estimates which are comparable to those obtained by more labour-intense (and often more invasive) capture-mark-recapture approaches, and because explanatory variables that may influence $\lambda$ and *p* can be investigated in a straightforward way using generalized linear models (GLMs; *Courtois et al., 2016*; *Ficetola et al., 2018*; *Priol et al., 2014*). *N*-mixture models have already been applied to a wide range of wildlife species (*e.g. Belant et al., 2016*; *Hunter, Nibbelink & Cooper, 2017*; *Kéry, 2018*; *Kidwai et al., 2019*; *Manica et al., 2019*; *Romano et al., 2017*; *Ward et al., 2017*), but are still considered as an emerging framework with ongoing extensions to original parameterizations (*Barker et al., 2018*; *Bötsch, Jenni & Kéry, 2019*; *Denes, Silveira & Beissinger, 2015*; *Kéry & Royle, 2016*).

Despite their large size, crocodylians are an example taxonomic group for which imperfect detection during surveys is common (*e.g. Balaguera-Reina et al., 2018*; *Da Silveira, Magnusson & Thorbjarnarson, 2008*; *Hutton & Woolhouse, 1989*). Historically, crocodylian population size estimations outside the capture-recapture framework have largely been based on spotlight surveys to reveal minimal counts, or counts accounting for visible fractions (*Balaguera-Reina et al., 2018*; *Bayliss, 1987*; *King, Espinal & Cerrato, 1990*). Although not yet widely used, *N*-mixture models have already been explored to investigate the influence of covariates on both local abundance and crocodylian detection (*Cartagena-Otálvaro et al., 2020*; *Farris et al., 2021*; *Fujisaki et al., 2011*; *Gardner et al., 2016*; *Lyet et al., 2016*; *Mazzotti et al., 2019*; *Naveda-Rodriguez, Utreras & Zapata-Ríos, 2020*; *Than et al., 2020*; *Waddle et al., 2015*), but detailed comparisons with more traditional methods particularly in hydrologically dynamic habitats are as yet lacking.

The Morelet's crocodile (*Crocodylus moreletii*) is a medium-to-large crocodile species occurring in Atlantic lowlands surrounding the Gulf of Mexico (Belize, Guatemala and Mexico; *Sigler & Gallegos, 2017*; Fig. 1). Our knowledge on the population ecology and status of *C. moreletii* has markedly increased over the last decades, and a standard international survey program to monitor its wild populations was developed in 2010 (*Sánchez-Herrera et al., 2011*). However, only rudimentary information about this species is available for the southern region of the Yucatan Peninsula, an area which is characterized by very dynamic hydrological regimes and harbours parts of the largest remaining expanse of tropical forest in Mesoamerica (*Carr, 1999*; *Vester et al., 2007*). In the

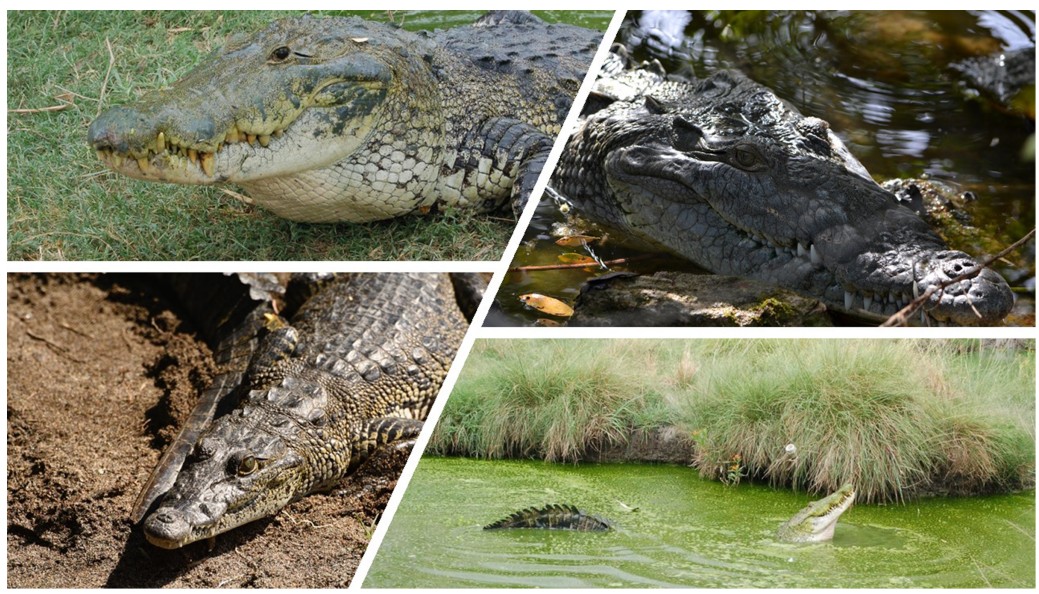

**Figure 1  A selection of individuals of Morelet's crocodile (*Crocodylus moreletii*) from the study area in Calakmul Biosphere Reserve in Mexico.**

present study, we estimate both local abundances as well as the total population size of *C. moreletii* in this region, using a set of binomial *N*-mixture modelling approaches for comparison with more traditionally used methods. Because *C. moreletii* locally inhabits particularly unstable and heterogeneous waterbodies, the study area provides an excellent opportunity to probe the versatility of *N*-mixture models under highly variable levels of detectabilities.

## MATERIALS AND METHODS

### Study area and data collection

Calakmul Biosphere Reserve (CBR) is located within the southern portion of the Yucatan Peninsula in Campeche, México (18°21.921′N, 089°53.220′W; Fig. 2), and together with the adjacent state reserves Balam-Ku and Balam-Kin encompasses more than 1.2 million hectares of protected forest for which *C. moreletii* represents one of the main flagship species. CBR is part of the Selva Maya, which was home to the ancient Mayan civilization and covers 10.6 million hectares of forest across Mexico, Guatemala and Belize (*Vester et al., 2007*). Precipitation gradually increases from 900 mm annually in the north to 1,400 mm in the south, with significant effects on local forest structure and tree species composition (*Martínez & Galindo-Leal, 2002*; *Vester et al., 2007*). The majority of the CBR is composed of tropical semi-deciduous forest with a canopy ranging from 15 to 40 m in height, with the northern parts containing deciduous forest with canopy heights of 8 to 20 m (*Chowdhury, 2006*). The geological characteristics of the CBR result in rapid rainwater belowground runoff, and non-permanent as well as semi-permanent small to medium-sized waterbodies, locally known as aguadas, represent the only source of water during the dry season (*Barão-Nóbrega, 2019*; *García-Gil, Palacio-Prieto & Ortiz-Pérez,*

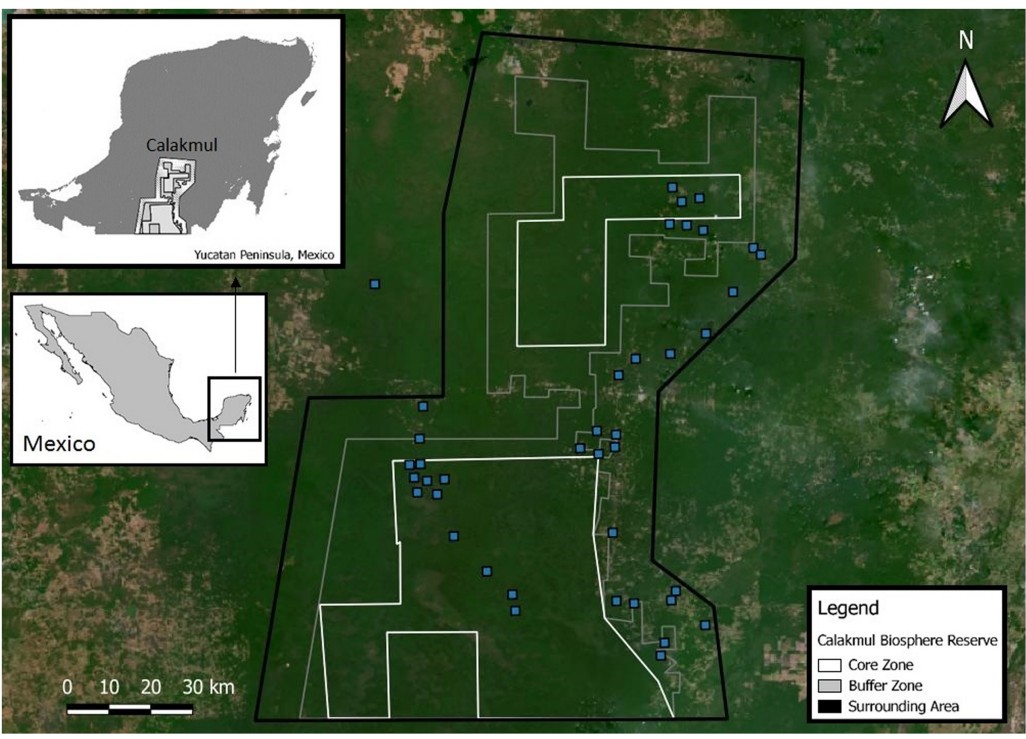

**Figure 2 Location of the Calakmul Biosphere Reserve (CBR) in southern-central region of the Yucatan Peninsula (Mexico).** The area delimited inside the inner lines within the grey and dark grey areas (CBR) represent, respectively, the politically established buffer and core zones of the biosphere reserve. Blue squares represent *Crocodylus moreletii* survey locations.

*2002*; *Reyna-Hurtado et al., 2010*). The distribution, prevalence, and morphology of aguadas across Calakmul is strongly influenced by annual precipitation cycles, resulting in high levels of seasonal and yearly variation in their general structure (hydric coverage, vegetation communities and cover; *Barão-Nóbrega, 2019*; *García-Gil, Palacio-Prieto & Ortiz-Pérez, 2002*; *Márdero et al., 2019*).

Candidate aguadas for *C. moreletii* surveys were identified using existing information (*García-Gil, 2000*), satellite imagery in *Google Earth Pro* (*Gorelick et al., 2017*), and local knowledge by environmental authorities (*Comisión Nacional de Áreas Naturales Protegidas*—CONANP and *Pronatura Península Yucatán*), local guides and community representatives to define a total of 40 survey locations spread across the CBR and its surroundings. While accessibility by vehicles as well as landowner permissions were a prerequisite for field surveys, information on the presence or absence of *C. moreletii* was largely unavailable for survey site selection. Exhaustive nocturnal spotlight counts were performed in July 2017 and in March as well as July 2018–2019 in all 40 waterbodies whenever possible (due to logistical reasons each waterbody was visited only once within a surveying campaign, *e.g.* July 2018). Vegetation cover and hydric state (relation between current water surface and maximum flooding capacity) in all waterbodies during each visit were visually estimated (for details see *Barão-Nóbrega, 2019*; Fig. S1), and classified as following: Vegetation Cover—Low (0–30%), Moderate (30–60%) or High (>60%); Hydric

State–Dry, Drying (0–25%), Stable (25–75%) or Full (>75%). We did not consider publicly available governmental precipitation data for the region, as they contained long periods of unavailable information, and do not locally correlate with observed aguada water levels (*Reyna-Hurtado et al., 2019*).

Avoiding days of full moon, high winds and heavy rain, surveys were conducted by systematically traveling along the perimeter of the waterbody on foot or by paddling along the shoreline aboard a 3.5 m aluminium boat. To ensure no section of the waterbody was overlooked, spotlighting was always carried using a frontal cone spotlight search radius of between 0° to 90° (*Sánchez-Herrera et al., 2011*). Individual *C. moreletii* were located by their eyeshine reflection, and, when detected, classified by size (hatchling, juvenile, sub-adult, adult) based on head length (*Sánchez-Herrera et al., 2011*). Crocodile wariness and flight distances across our study sites were low, which enabled unambiguous designations of size classes. If the waterbody was dry, the number of detected crocodiles was assumed zero. Crocodile hatchlings (TL < 30 cm) have high mortality rates (*Grigg & Kirshner, 2015*) and were excluded from the count data (*Balaguera-Reina et al., 2018*). Research permits for fieldwork activities in CBR were issued annually by Mexico's Secretariat of Environment and Natural Resources (SEMARNAT; SGPA/DGVS/ 03030/17; SGPA/DGVS/005403/18) and National Commission of Natural Protected Areas (CONANP; D-RBC-118/2017; D-RBC-030/2018; D-RBC-087/2019).

## Abundance estimation

The most common spatial unit used in crocodylian studies is number of crocodiles per surveyed kilometre (*Sánchez-Herrera et al., 2011*), appliable to surveying routes which span across *e.g.* rivers and large lakes. In the region of Calakmul, for the majority of the year aquatic habitat largely consists of small to medium size aguadas (<1 ha; *Barão-Nóbrega, 2019*; *García-Gil, 2000*; *García-Gil, Palacio-Prieto & Ortiz-Pérez, 2002*) that occur at relatively low densities across the region (on average less than one non-dry waterbody per 535 hectares, *Delgado-Martínez & Mendoza, 2020*; *García-Gil, 2000*). We therefore refrained from using kilometres as the spatial unit for abundance and focused on the number of individuals per aguada relative to aguada perimeter as a more informative level of investigation.

As abundance estimation methods which are in common use for many crocodylians including *C. moreletii* (*Cedeño-Vázquez, Ross & Calme, 2006*; *Tellez et al., 2017*), we used unadjusted count data (Minimum Population Size—MPS) as well as the King's Visible Fraction Method (VFM). MPS quantifies the number of recorded individuals per spatial unit (in our case, survey site) to obtain baseline data (*Sánchez-Herrera et al., 2011*), and represents the minimum number of animals in a stable population (*Hutton & Woolhouse, 1989*). In our study, the highest crocodile count value observed between 2017–2019 in each surveyed location was used as representative of its local MPS abundance ($n^{MPS}$). Due to the topography of our studied region coupled with the spatial distribution of surveyed sites, we assume that during the 3-year period (2017–2019) there was no movement of individuals between surveyed locations, and that differences (albeit non-significant; Friedman chi-squared = 1.9632, df = 4, *p*-value = 0.7425; Fig. S2) in crocodile

counts within the same location between surveying periods were due to variation in our ability to detect crocodiles under different environmental conditions (vegetation cover and water level).

VFM makes use of repeated counts per site, estimating the percentage of the total population observed during a single count (the visible fraction) as $vf = \frac{\bar{x}}{1.05(2\sigma + \bar{x})}$, where σ is the standard deviation and $\bar{x}$ is the average number of crocodiles counted (*Balaguera-Reina et al., 2018*; *King, Espinal & Cerrato, 1990*). This equation attempts to estimate the unknown relationship between the observed and real values of abundance. Local abundances ($n^{\text{VFM}}$) at given waterbodies can therefore be expressed as $n^{\text{VFM}} = \frac{\bar{x}}{vf} \pm \frac{[1.96(\sigma)]^{1/2}}{vf}$, with a 95% confidence interval (CI). Although the latter method has brought important insights regarding the ecology of some crocodylians, meeting the statistical assumptions to estimate sighting fractions is not easy to achieve in natural populations (see *Balaguera-Reina et al., 2018* for further discussion).

In addition to these traditionally applied methods, we focused on binomial *N*-mixture models (*Royle, 2004*; hereafter referred to as "*N*-mixture models") and explored three alternative statistical distributions: Poisson, Negative Binomial (NB), and Zero-Inflated Poisson (ZIP). Poisson distributions are generally applied to describe relative density but have a variance which is equal to its mean and therefore do not conform well to under- or over-dispersed data (*Denes, Silveira & Beissinger, 2015*). Both Poisson and NB distributions are usually adequate to model abundance when only considering occupied survey sites, but tend to perform poorly in the presence of a significant number of true zeros in the dataset (*Joseph et al., 2009*; *Wenger & Freeman, 2008*), whereas the ZIP distribution is generally able to better accommodate both true and false zeros (*Denes, Silveira & Beissinger, 2015*). Both ZIP and NB distributions allow for over-dispersion, a common feature of population count data (*Knape et al., 2018*), but only to a certain point (*Kéry & Royle, 2016*; *Warton et al., 2017*).

Sixteen *N*-mixture models per distribution (Poisson, NB and ZIP) were fitted to the dataset using the *pcount()* function in the *unmarked* package in R (*Fiske & Chandler, 2011*; *R Development Core Team, 2019*), using *R*Studio version 1.1.456 (*RStudio Team, 2016*). Waterbody Perimeter (in metres) was used as a covariate for λ, in addition to a Geographic Specifier (*i.e.*, in which surveyed region within Calakmul was the waterbody located) to account for possible spatial differences in abundance within Calakmul. Vegetation Cover and waterbody Hydric State were used as covariates of *p*. *N*-mixture modelling as implemented in the *unmarked* package for R (*Fiske & Chandler, 2011*) approximates the likelihood by truncating an infinite sum over all possible values of abundance. An integer value specifying the upper index of integration (*K*) needs to be set when fitting the model, but estimates can be unstable to changes in this bound, possibly due to maximum likelihood estimates of abundance being infinite (*Dennis, Morgan & Ridout, 2015*; *Kéry & Royle, 2016*; *Knape et al., 2018*). We used a numeric upper bound *K* = 150 for abundance in the calculation of the likelihoods after trials determined that this was large

enough to have no effect on the model results (*Fiske & Chandler, 2011*). Models that failed to converge were discarded, and the Akaike's Information Criterion (AIC, *Akaike, 1974*) was used to identify the best models within each *N*-mixture structure (Poisson, NB and ZIP).

Parametric bootstrapping (100 simulations) was conducted using the *parboot*() function (*Fiske & Chandler, 2011*) to calculate *p*-values from sums of squares (SSE), Pearson's Chi-square and Freeman–Tukey fit statistics that quantified the fit of the best model to the dataset within each *N*-mixture structure. A dispersion parameter (ĉ) was calculated as the ratio of the observed fit statistic to the mean of the simulated distribution (*Kéry & Royle, 2016*). As caution is advised when using NB even when it is greatly favoured by AIC, particularly when this distribution produces substantially higher estimates than Poisson and ZIP distributions (*Kéry, 2018*; *Kéry & Royle, 2016*), we further investigated which model structure would provide the highest overall confidence in crocodile abundance estimations by running residual diagnostic analyses using the *plot.Nmix.resi*() and *residqq*() functions, available in the *AHMbook* (*Kéry & Royle, 2016*) and *nmixgof* (*Knape et al., 2018*) packages for *R*. The benefit over the bootstrapping procedure (*i.e.* the *parboot*() function) is that these two approaches take significantly less time to compute, and results can be used to graphically check a range of assumptions such as overdispersion *via* quantile–quantile plots (qq plots) and residual plots against fitted values to assess homoscedasticity (*Knape et al., 2018*; *Warton et al., 2017*). Once the best overall model was selected, the *predict*() function in *unmarked* was used to estimate local abundances ($n^{N\text{-mixture}}$) at surveyed waterbodies. This *predict*() function was also used to generate estimated relationships with predictors for each covariate of $\lambda$ and $p$, which were then plotted using the plotting functions *ggplot2* package for *R* (*Wickham, 2016*).

To estimate total population size ($\hat{N}$) across the entire study region irrespective of surveyed waterbodies, we extrapolated the relationship between the three crocodile abundance estimators used ($n^{MPS}$, $n^{VFM}$ and $n^{N\text{-mixture}}$) and waterbody perimeter to all aguadas known to occur in studied region (CBR and surroundings). To achieve this, vector information on all semi-temporary and permanent aguadas was extracted from an existing hydrological GIS-based dataset (*García-Gil, 2000*), and their perimeters were estimated using QGIS Desktop software version 3.0.2 (*QGIS Development Team, 2019*). For $n^{N\text{-mixture}}$, the *predict*() function was used to generate estimates of abundance for all waterbodies based on the previously generated relationship between $\lambda$ and waterbody perimeter. For $n^{MPS}$ and $n^{VFM}$, the relationship between local *C. moreletii* abundance and the perimeter of each surveyed waterbody was determined through a Generalized Linear Model (abundance perimeter~family = poisson) using the *glm*() function in *R*, then extrapolated to all waterbodies using the *predict*() function. For additional reference, total population size ($\hat{N}$) based on $n^{MPS}$, $n^{VFM}$ and $n^{N\text{-mixture}}$ was further estimated by extrapolating the observed relationship between the number of crocodiles in the combined surveyed perimeter to the total perimeter in the entire study region of Calakmul:

$$\frac{Number\ of\ crocodiles\ -\ Total\ surveyed\ perimeter}{\hat{N}\ -\ Total\ perimeter\ in\ the\ studied\ region}.$$

## RESULTS

A total of 191 spotlight surveys in 40 waterbodies were conducted between 2017 and 2019 (34 surveys in July 2017, 40 and 38 surveys in March and July 2018, as well as 39 and 40 surveys in March and July 2019, respectively), yielding a combined total of 905 *C. moreletii* detections (average and total waterbody perimeters surveyed: 450 and 18,010 m, respectively). In all 40 waterbodies, between four and five spotlight surveys were carried out over the entire studied period. *Crocodylus moreletii* was detected at least once in 21 of the 40 surveyed sites (52%). Maximum number of crocodiles counted per waterbody ($n^{MPS}$) ranged between 0 and 89 (median = 1; $Q_1$–$Q_3$ = 0–7). Across sites, *vf* varied between 0.18 and 0.95 (mean ± SD = 0.48 ± 0.25), with the resulting local population size estimates ($n^{VFM}$) ranging from 0 to 112 individuals (median = 1; $Q_1$–$Q_3$ = 0–9).

Considering all possible combinations of covariates, 16 *N*-mixture models were generated for each distribution (Poisson, ZIP and NB). The full model [*p* (Vegetation Cover + Hydric State), λ (waterbody Perimeter + Geographic Specifier)], which accounts for waterbody location and perimeter on λ as well as the cumulative effects between co-variates on *p*, exhibited the lowest AIC values amongst all possible combinations (Table 1). Local abundances ($n^{N\text{-}mixture}$) generated through this model fitted to each distribution ranged from zero to 140 ± 108 (median = 2 ± 1; $Q_1$ = 0; $Q_3$ = 16 ± 8) individuals per waterbody when using NB, from zero to 138 ± 16 (median = 3 ± 1; $Q_1$ = 0; $Q_3$ = 18 ± 4) when using Poisson, and from zero to 120 ± 18 (median = 6 ± 3; $Q_1$ = 0; $Q_3$ = 18 ± 4) when using ZIP.

Despite the NB distribution yielding the lowest overall AIC value, it showed poor residual diagnostic performance whereas the Poisson and ZIP models achieved a better agreement between both observed and expected data as well as between residuals and fitted values (Table 1; Fig. 3). The qq plots of site-sum randomized quantile residuals indicate that the model fitted with Poisson distribution also provides a poor fit to the data since the quantiles deviate clearly from the identity line (Fig. S3). Despite the qq plots of sit-sum for NB and ZIP exhibiting a relatively similar behaviour (Fig. S3), the former exhibited poor residual diagnostic performance and wide-ranging confidence intervals (Fig. 3; Fig. S4). Goodness-of-fit bootstrap statistics also favoured the best-fit ZIP *N*-mixture model (sum of squares—SSE: *p* = 0.32; Pearson's Chi-square: *p* = 0.17; Freeman–Tukey: *p* = 0.46), suggesting a more acceptable fit to the data. The value of $\hat{c}$ (ratio of observed/expected) for this *N*-mixture ZIP model was 1.28, indicating only slight over-dispersion. Taken together, we considered that *N*-mixture ZIP provided superior model confidence over the Poisson and NB models.

Based on the best fit *N*-mixture ZIP model, the highest probabilities of detection were observed in Drying waterbodies with Low (*p* = 0.68) to Moderate vegetation cover (*p* = 0.62), whereas lowest detectability was associated with High vegetation cover (*p* < 0.45, Fig. 4). The relationships between waterbody perimeter and local abundance estimates ($n^{MPS}$, $n^{VFM}$ and $n^{N\text{-}mixture}$; Fig. 5) were used to provide an estimate for the total number of non-hatchling *C. moreletii* across the study area (Table 2). Analyses of the
**Table 1 Abundance estimation models of *Crocodylus moreletii* in Calakmul using three different N-mixture model approaches (Poisson, Negative Binomial—NB and Zero Inflated Poisson—ZIP).** Models were fitted with different combinations of waterbody perimeter as covariate of abundance ($\lambda$) and vegetation cover (Low, Moderate, High) and water level (Dry, Drying, Stable, Full) as categorical co-variables of detection ($p$). The models highlighted in bold exhibited the lowest AIC values amongst all possible N-mixture combinations.

| Model structure | Model covariates | AIC | (–) LogLike | ΔAIC | AIC weight |
|---|---|---|---|---|---|
| Poisson | **$p$ (Water + Veg), $\lambda$ (Perimeter + Gsp)** | **473** | **218** | **0** | **>0.99** |
| | $p$ (Water), $\lambda$ (Perimeter + Gsp) | 493 | 230 | 20 | <0.01 |
| | $p$ (Water + Veg), $\lambda$ (Gsp) | 524 | 245 | 51 | <0.01 |
| | $p$ (Water), $\lambda$ (Gsp) | 543 | 256 | 70 | <0.01 |
| | $p$ (Veg), $\lambda$ (Perimeter + Gsp) | 680 | 325 | 207 | <0.01 |
| | $p$ (Water + Veg), $\lambda$ (Perimeter) | 737 | 360 | 264 | <0.01 |
| | $p$ (Veg), $\lambda$ (Gsp) | 747 | 359 | 274 | <0.01 |
| | $p$ (Water), $\lambda$ (Perimeter) | 765 | 376 | 291 | <0.01 |
| | $p$ (.), $\lambda$ (Perimeter + Gsp) | 902 | 438 | 429 | <0.01 |
| | $p$ (.), $\lambda$ (Gsp) | 968 | 472 | 495 | <0.01 |
| | $p$ (Veg), $\lambda$ (Perimeter) | 1,012 | 501 | 539 | <0.01 |
| | $p$ (Water + Veg), $\lambda$ (.) | 1,058 | 522 | 585 | <0.01 |
| | $p$ (Water), $\lambda$ (.) | 1,094 | 542 | 612 | <0.01 |
| | $p$ (.), $\lambda$ (Perimeter) | 1,282 | 638 | 809 | <0.01 |
| | $p$ (Veg), $\lambda$ (.) | 1,446 | 719 | 973 | <0.01 |
| | $p$ (.), $\lambda$ (.) | 1,710 | 853 | 1,237 | <0.01 |
| NB | **$p$ (Water + Veg), $\lambda$ (Perimeter + Gsp)** | **416** | **189** | **0** | **0.56** |
| | $p$ (Water), $\lambda$ (Perimeter + Gsp) | 417 | 191 | 1 | 0.43 |
| | $p$ (Water + Veg), $\lambda$ (Gsp) | 427 | 195 | 10 | <0.01 |
| | $p$ (Water), $\lambda$ (Gsp) | 427 | 197 | 10 | <0.01 |
| | $p$ (Water + Veg), $\lambda$ (Perimeter) | 434 | 208 | 18 | <0.01 |
| | $p$ (Water), $\lambda$ (Perimeter) | 434 | 210 | 18 | <0.01 |
| | $p$ (Water), $\lambda$ (.) | 439 | 213 | 23 | <0.01 |
| | $p$ (Water + Veg), $\lambda$ (.) | 440 | 212 | 24 | <0.01 |
| | $p$ (Veg), $\lambda$ (Perimeter + Gsp) | 603 | 285 | 186 | <0.01 |
| | $p$ (Veg), $\lambda$ (Gsp) | 614 | 292 | 197 | <0.01 |
| | $p$ (Veg), $\lambda$ (Perimeter) | 626 | 307 | 209 | <0.01 |
| | $p$ (Veg), $\lambda$ (.) | 633 | 311 | 216 | <0.01 |
| | $p$ (.), $\lambda$ (Perimeter + Gsp) | 728 | 350 | 311 | <0.01 |
| | $p$ (.), $\lambda$ (Gsp) | 737 | 355 | 320 | <0.01 |
| | $p$ (.), $\lambda$ (Perimeter) | 748 | 370 | 331 | <0.01 |
| | $p$ (.), $\lambda$ (.) | 742 | 373 | 335 | <0.01 |
| ZIP | **$p$ (Water + Veg), $\lambda$ (Perimeter + Gsp)** | **448** | **205** | **0** | **>0.99** |
| | $p$ (Water), $\lambda$ (Perimeter + Gsp) | 459 | 212 | 11 | <0.01 |
| | $p$ (Water + Veg), $\lambda$ (Gsp) | 483 | 223 | 35 | <0.01 |
| | $p$ (Water), $\lambda$ (Gsp) | 491 | 229 | 42 | <0.01 |
| | $p$ (Water + Veg), $\lambda$ (Perimeter) | 601 | 291 | 153 | <0.01 |
| | $p$ (Water), $\lambda$ (Perimeter) | 613 | 299 | 165 | <0.01 |

| Model structure | Model covariates | AIC | (–) LogLike | ΔAIC | AIC weight |
|---|---|---|---|---|---|
| | $p$ (Veg), $\lambda$ (Perimeter + Gsp) | 649 | 308 | 201 | <0.01 |
| | $p$ (Veg), $\lambda$ (Gsp) | 699 | 334 | 251 | <0.01 |
| | $p$ (Water + Veg), $\lambda$ (.) | 790 | 387 | 341 | <0.01 |
| | $p$ (Water), $\lambda$ (.) | 797 | 392 | 349 | <0.01 |
| | $p$ (Veg), $\lambda$ (Perimeter) | 820 | 404 | 371 | <0.01 |
| | $p$ (.), $\lambda$ (Perimeter + Gsp) | 859 | 415 | 411 | <0.01 |
| | $p$ (.), $\lambda$ (Gsp) | 905 | 439 | 456 | <0.01 |
| | $p$ (.), $\lambda$ (Perimeter) | 1,040 | 516 | 592 | <0.01 |
| | $p$ (Veg), $\lambda$ (.) | 1,066 | 528 | 617 | <0.01 |
| | $p$ (.), $\lambda$ (.) | 1,268 | 631 | 819 | <0.01 |

GIS-based dataset revealed a total of 1,663 aguadas, which had a median perimeter of 139 m (min = 40 m; max = 3,639 m; $Q_1$ = 98 m; $Q_3$ = 207 m; Fig. S5). Based on these numbers, MPS$^{GLM}$ estimated a total *C. moreletii* population size as 7,753 (7,190–8,361) non-hatchling individuals, which is slightly below the values obtained by approaches which take detectability into account (Table 2). The *N*-mixture ZIP model estimated a total population size of 10,440 crocodiles (9,800–11,123), compared to 9,619 (8,989–10,294) individuals obtained by VFM$^{GLM}$.

# DISCUSSION

Allowing for the separate estimation of abundance and detection probabilities from replicated counts of unmarked individuals (*e.g. Kéry & Royle, 2016*; *Zipkin et al., 2014*), *N*-mixture models have in recent years become applied to taxa ranging from mosquitoes to megafaunal mammals (*Kidwai et al., 2019*; *Manica et al., 2019*). In the present study, we applied a set of such models to spotlight count data for *C. moreletii* in southern Yucatan, where it inhabits particularly dynamic waterbodies and serves as an important flagship species for a large expanse of protected forest. We argue that *N*-mixture modelling applied to our highly heterogeneous dataset provided the most insightful estimates of crocodile abundance and detection in our study region. We also generally show that *N*-mixture models offer a flexible approach for abundance estimates when ecological conditions cause wide variations in detectability, and provide the first estimate for total population size across the CBR. Although direct comparisons with other study areas are hampered by structurally highly dissimilar survey environments (*i.e.* relatively small semi-temporary ponds *vs.* large lakes, lagoons and rivers), we suggest that the CBR represents an important stronghold for *C. moreletii* because the estimated population size for our study area currently represents about 7–13% of the total population assumed for the entire country of Mexico (which itself represents the majority of the *C. moreletii* range, with only peripheral further populations occurring in Belize and Guatemala).

Unadjusted count data are often used as a proxy for true abundance due to the challenges involved in the monitoring of wildlife populations (*Farris et al., 2021*; *Gerber &*

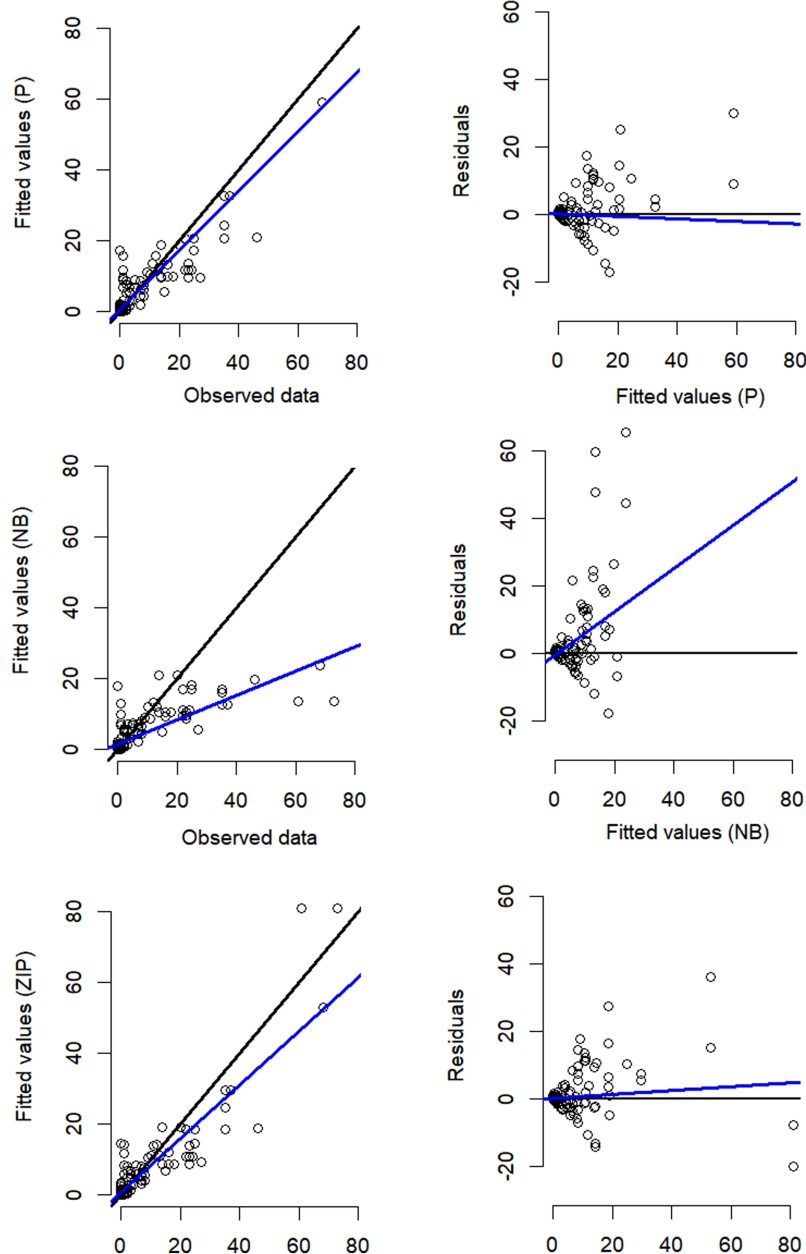

**Figure 3 Residual diagnostics for the best model of each of the three *N*-mixture modelling approaches fitted to the *Crocodylus moreletii* spotlight count dataset.** Left hand side figures represent Poisson (P), Negative Binomial (NB) and Zero Inflated Poisson (ZIP) *N*-mixture fitted values *vs.* observed crocodile counts, where the black line shows a 1:1 relationship and the blue line is the linear regression line of best fit. Right hand side figures represent residuals *vs.* fitted values (black line denotes a zero residual and the blue line is the linear regression line).

*Kendall, 2017*; *Johnson, 2008*). However, the implicit assumption that the relationship between observed counts and actual population sizes remains constant is often violated due to varying levels of detectability (*Balaguera-Reina et al., 2018*; *Gerber & Kendall, 2017*; *Kéry, Royle & Schmid, 2005*; *Kéry & Schmidt, 2008*). For crocodylians, count data may indeed serve as abundance surrogates to capture temporal and spatial population trends
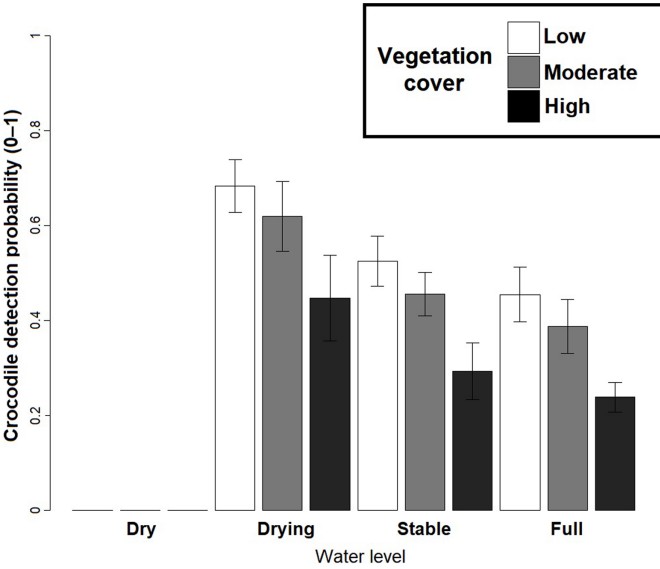

**Figure 4** *Crocodylus moreletii* **detection probability estimations in function of water level and vegetation cover inside the waterbody calculated through Zero Inflated Poisson *N*-mixture modelling.**

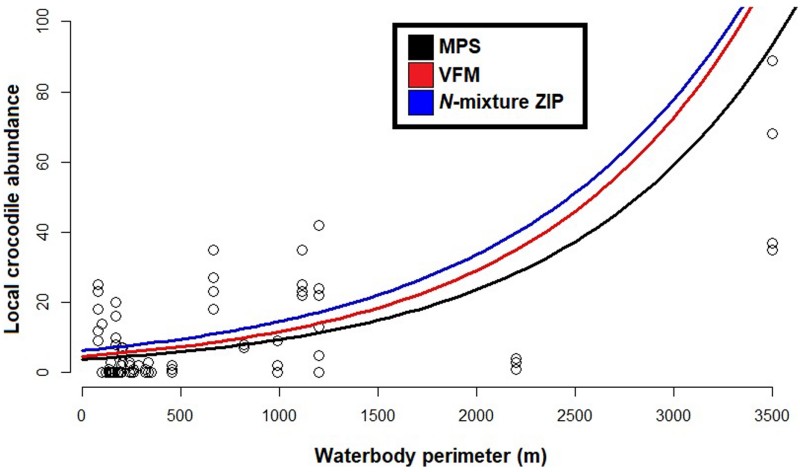

**Figure 5 Generalized linear models between local *Crocodylus moreletii* abundance and waterbody perimeter in the region of Calakmul.** Red and black lines represent, respecively, fitted values from estimations through King's visible fraction method (VFM) and Minimum Population Size (MPS) using only baseline count data. Dark blue line represents fitted values calculated through Zero Inflated Poisson *N*-mixture modelling. Open circles represent all our baseline crocodile count values (*i.e.* number crocodiles observed during the survey).

related with for example habitat structure or human induced changes to the environment, but is best used when conditions during multiple surveys allow for the recording of similar proportions of individuals (*Bayliss, 1987*; *Platt & Thorbjarnarson, 2000*). Unadjusted spotlight count data could lead to underestimated population sizes due to potential factors of bias in the environment (*e.g.* vegetation density, crocodile wariness) that prevent detection of all animals present during a site visit (*Bayliss et al., 1986*; *Cherkiss, Mazzotti &*

**Table 2 Estimates of total population size of *Crocodylus moreletii* occurring in the region of Calakmul based on five different estimation approaches.**

| Approach | Total population size |
| --- | --- |
| MPS[GLM] | 7,753 (7,190–8,361) |
| VFM[GLM] | 9,619 (8,989–10,294) |
| *N*-Mixture ZIP | 10,440 (9,800–11,123) |
| MPS[3] | 5,567 |
| VFM[3] | 6,884 (5,002–9,224) |
| *N*-Mixture ZIP[3] | 8,322 (6,538–10,105) |

Note:
MPS, crocodile count data alone (without considering detection probability); VFM, King's visible fraction method; MPS[GLM], Generalized linear model considering the relation between MPS and waterbody perimeter; VFM[GLM], Generalized linear model considering the relation between VFM and waterbody perimeter; ZIP, Zero Inflated Poisson *N*-mixture approach. [GLM] represents total population size calculated through generalized linear model considering the relation between local abundance (MPS or VFM) and waterbody perimeter. [3] Total population size calculated thorough the application of a rule of three: Range inside parentheses represent the lower and upper prediction range within a 95% confidence for total abundance of crocodiles in Calakmul.

*Rice, 2006*; *Nair et al., 2012*). Because many of these factors can be standardized to minimize their effects, unadjusted counts as an index of abundance remain commonly applied (*Balaguera-Reina et al., 2018*; *Farris et al., 2021*; *Hutton & Woolhouse, 1989*; *Sánchez-Herrera et al., 2011*).

Our study revealed no significant difference between number of crocodiles counted between surveying campaigns, and we therefore assumed that local abundances overall remained constant during the 3-year study period, precluding temporal variation of λ. We acknowledge that the assumption of closed populations fails to take demographic turnover (*e.g.* mortality) and migration to and from also unsurveyed locations into account (for a general discussion see *e.g. Royle & Dorazio, 2006*). Logistic reasons however precluded spotlight count replicates for each site within the five surveying campaigns, with single-visit surveys in multisite areas being common for long-lived study organisms such as crocodylians (*Balaguera-Reina et al., 2018*; *Cartagena-Otálvaro et al., 2020*; *Farris et al., 2021*; *Hutton & Woolhouse, 1989*; *Marioni, Von Muhlen & Da Silveira, 2007*; see also the Mexican National Monitoring Program for *C. moreletii*, *Sánchez-Herrera et al., 2011*).

Due to an evident effect on these variables on surveys, water level and vegetation cover were considered as covariates of detection (Fig. 4). In Calakmul, decreasing water availability caused by disruptions in the timing and intensity of local precipitation resulted in marked shifts in water level and vegetation cover of aguadas across the study period (*Barão-Nóbrega, 2019*; *Márdero et al., 2019*; *Reyna-Hurtado et al., 2019*). Under such conditions, the observer's ability to detect individuals will widely vary both temporally as well as spatially, requiring the effects of environmental conditions to be taken into account for obtaining accurate estimates (see also *Fujisaki et al., 2011*).

Given that the majority of *C. moreletii* habitat across its range is represented by rivers and lakes where spotlight searches are conducted over larger areas than in our more confined aguada habitats (*Cedeño-Vázquez, Ross & Calme, 2006*; *Sánchez-Herrera et al., 2011*; *Tellez et al., 2017*), we argue that our surveys provide particularly accurate information. The non-linear increase of abundances with increasing size of aguadas for

the MPS, VFM and *N*-mixture ZIP curves is likely linked to larger waterbodies representing more hydrologically stable environments, therefore supporting higher relative numbers of reproductive individuals (*Sánchez-Herrera et al., 2011*). Small to medium sized aguadas are generally more prone to desiccation (*Barão-Nóbrega, 2019*), and are often inhabited by only 1–2 adult *C. moreletii* which might not locally reproduce (*Barão-Nóbrega, 2021*). It also needs to be born in mind that the relationship between waterbody surface and volume with perimeter, while depending on the overall shape, is non-linear in general (*García-Gil, Palacio-Prieto & Ortiz-Pérez, 2002*; *Gunn et al., 2002*). Given that detection rates are not accounted for in the MPS approach, its lower abundance values are not surprising (*Balaguera-Reina et al., 2018*; *Farris et al., 2021*; *MacKenzie et al., 2003*; *Naveda-Rodriguez, Utreras & Zapata-Ríos, 2020*). The VFM and the *N*-mixture ZIP model, which on the other hand consider detectability, estimate higher abundance values than MPS. For waterbodies below 1,200 m in perimeter, a size class which accounts for 98% of aguadas in the study area (*Barão-Nóbrega, 2019*; *García-Gil, Palacio-Prieto & Ortiz-Pérez, 2002*) this difference is less pronounced than for larger waterbodies (Fig. 5); that surface area is related to encounter rate and detection for crocodylians has been previously documented (*e.g. Da Silveira, Magnusson & Thorbjarnarson, 2008*; *Fujisaki et al., 2011*).

Estimates of visible fractions obtained through VFM and detection probabilities obtained through *N*-mixture modelling were naturally different, which is also reflected in the higher abundance estimates as predicted by the *N*-mixture model in comparison with VFM across the entire perimeter range. Part of the reason for this difference might be related to the former method estimating the proportion of all members of the population that were detected (*Balaguera-Reina et al., 2018*; *Cartagena-Otálvaro et al., 2020*; *Morales-Betancourt et al., 2013*) while the latter is based on individual detection probability (*Kéry & Royle, 2016*). Whilst not the case in our study, significantly lower detection probability values when compared with visible fractions has been reported in two studies involving crocodylian count data and suggested to be linked to small sample sizes (*Cartagena-Otálvaro et al., 2020*; *Mazzotti et al., 2019*). *N*-mixture modelling is best used on datasets containing substantial sample sizes (multiple observations and/or multiple survey sites), which provide increased power to statistical analysis and allow for higher confidence in estimating population sizes and detection probabilities, along with estimation of spatial and temporal variation associated with them (*Cartagena-Otálvaro et al., 2020*; *Couturier et al., 2013*; *Dennis, Morgan & Ridout, 2015*; *Keever et al., 2017*; *Kéry & Royle, 2016*; *Yamaura, Kéry & Andrew Royle, 2016*).

*N*-mixture models yield unbiased estimates of abundance and detectability in simulated datasets of closed populations (*Kéry & Royle, 2016*; *Royle, 2004*), but benchmarks to assess the performance of *N*-mixture models from field data are difficult to obtain (*Kéry, Royle & Schmid, 2005*). Detectability is hardly ever equal to one, which means that field data can contain an unknown proportion of "false zeroes" (*i.e.* recorded absences of a species, but in reality the species was present but missed during a survey; *Bötsch, Jenni & Kéry, 2019*; *Della Rocca et al., 2020*; *Kéry & Royle, 2016*; *Kéry & Schmidt, 2008*). A particular feature of our dataset is a wide range of count values across sites, with zero

detections being a common occurrence (no *C. moreletii* were recorded for about 40% of aguadas, and approximately 60% of all surveys yielded in no counts). This likely led to the limited fit of our data to assumptions of specific distributions (*e.g.* negative binomial), and also to the apparent over-dispersion, which is a common problem in count data (*Kéry & Royle, 2016*; *Lee & Nelder, 2000*; *Ver Hoef & Boveng, 2007*). Estimates of local abundances in Drying aguadas were also possibly slightly biased downward, as such situations can lead to refugee behaviour inside dens and burrows within or in the vicinity of waterbodies (*Barão-Nóbrega et al., 2016*; *Platt, 2000*), as the risk of predation (*e.g.* by jaguar; *Panthera onca*) increases particularly for juveniles (*Sima-Pantí et al., 2020*). That the geographic specifier (*i.e.*, in which surveyed region within Calakmul was the waterbody located) significantly improved the performance of the *N*-mixture models however also suggests a high degree of site fidelity, supporting our assumption of stable populations within and between surveying campaigns; low levels of dispersal are also evidenced by genetic data on the relatedness structure within and between aguadas (J.A.L. Barão-Nóbrega et al., 2020, unpublished data). A particular strength of the *N*-mixture models was their ability to directly relate detectability with ecological parameters. While our findings confirm existing studies on the general nature of such relationships (*Bayliss et al., 1986*; *Cherkiss, Mazzotti & Rice, 2006*; *Da Silveira, Magnusson & Thorbjarnarson, 2008*; *Fujisaki et al., 2011*; *Montague, 1983*; *Wood et al., 1985*), they enabled an accurate quantification for the estimation and interpretation of *C. moreletii* abundances specifically for our study setting.

Amongst the three *N*-mixture model structures used in this study, the NB distribution overall exhibited lower AIC values but performed poorly during the residual diagnostic analysis and revealed excessively large confidence intervals, which could be linked to model unidentifiability (the "good fit/bad prediction dilemma"; see *Dennis, Morgan & Ridout, 2015*; *Joseph et al., 2009*; *Kéry, 2018*; *Kéry & Royle, 2016* for detailed discussion on this topic). NB-based models are also known to produce abundance estimates that may seem unrealistically high in comparison to values obtained through Poisson and ZIP, which has been linked to low projected detection probabilities (*e.g.* see *Cartagena-Otálvaro et al., 2020*; *Mazzotti et al., 2019*). We believe that, despite the overall lower AIC values exhibited by the NB model structure, the presence of two 'red flags' (wide confidence intervals and poor residual diagnostic performance) was sufficient for discarding NB in favour of ZIP, which performed better in this regard despite overall slightly higher AIC values. Hierarchical modelling of abundance from unmarked individuals using *N*-mixture models will remain a rich ground for both theoretical and applied investigations in the future (*Bötsch, Jenni & Kéry, 2019*; *Kéry, 2018*; *Kéry & Royle, 2016*).

Extrapolating our local abundance data across the entire region of Calakmul requires that surveyed aguadas are unbiased representatives for the entire area. While a randomization process for site selection was not possible due to logistic constraints (landowner permission and site accessibility), we did not take previous information on the presence or absence of *C. moreletii* into account and based our inferences on a large sample size of sites. Comparing our overall population sizes derived for Calakmul with country-wide estimates for *C. moreletii* numbers (largely based on the MPS approach,

the total population size in Mexico has been estimated at 78,157–104,815 individuals; *Álvarez, 2005*; *Rivera-Téllez et al., 2017*) reveals that the overall population size for our study area represents about 7–13% of the total population previously estimated for Mexico, highlighting Calakmul as an important stronghold for the conservation of *C. moreletii*. Furthermore, recent genetic data indicate that Calakmul still harbours non-admixed *C. moreletii* individuals (J.A.L. Barão-Nóbrega et al., unpublished data), reinforcing the importance of this region given that hybridization with the American crocodile *C. acutus* is common across other parts of its range (*Pacheco-Sierra et al., 2018*).

We believe that *N*-mixture models provide an overall comprehensive framework for the monitoring and management of crocodylian populations in general, as abundance and detection probability can be accurately estimated also in situations involving a high degree of environmental variation. *N*-mixture approaches can provide a rich ground for investigations aiming to analyse information relevant to future conservation management plans and should be more commonly integrated in monitoring schemes, due to the versatility of these models to analyse count data and provide insight to population responses to environmental conditions (*Costa, Romano & Salvidio, 2020*; *Duarte, Adams & Peterson, 2018*; *Kidwai et al., 2019*; *e.g.* see *Ward et al., 2017*). As for *C. moreletii*, the national monitoring program in Mexico (*Sánchez-Herrera et al., 2011*) should consider integrating a *N*-mixture modelling framework to analyse their survey data to better understand temporal and spatial population dynamic processes at both local and national scales. Furthermore, this monitoring program would also be a suitable testing ground for the creation of a user-friendly platform for *N*-mixture modelling analyses of abundance (similar to what has been done for example for distance sampling; http://distancesampling. org/), allowing a wider range of local managers to analyse their data in an easy to follow standardized system. One of the current deterrents for the use of *N*-mixture approaches is the amount of time and knowledge needed to carry out the analyses when using packages in *R* (*Fiske & Chandler, 2011*; *R Development Core Team, 2019*).

## CONCLUSIONS

Long-term monitoring data based on landscape-level systematic surveys provide useful information to describe spatial and temporal patterns of relative density in crocodylians (*Fujisaki et al., 2011*; *Waddle et al., 2015*). This study provides the first *C. moreletii* population size estimates for the south-central region of the Yucatan Peninsula, and reveals a large population of likely involving multiple reproduction areas across the region. More generally, the *N*-mixture models applied to spotlight count data have provided insightful estimates of crocodylian detection rates and local abundances in a particularly dynamic environment, enabling insights into population responses to local environmental conditions by allowing comparisons between highly heterogeneous survey sites. One of the caveats of our study was that, although the covariates used in this study allowed some understanding on the structured variation of the data, other unidentified variables are clearly affecting the distribution of our data. As such, future studies could expand the existing field surveys and *N*-mixture models to investigate whether further factors such as annual precipitation, water quality, surrounding forest structure, human activity and

reproductive activities account for local presence and abundance. Although not addressed in this study, changes in local abundance over time can be addressed by dynamic versions of this *N*-mixture modelling approach (*e.g.* the *pcountOpen*() function in unmarked). On a larger scale, we also recommend the use of *N*-mixture approaches to analyse existing and future *C. moreletii* spotlight count data across its range (*Álvarez, 2005*; *Rivera-Téllez et al., 2017*; *Sánchez-Herrera et al., 2011*), to provide more accurate baseline information for future conservation management plans at species level.

## ACKNOWLEDGEMENTS

We would like to thank the communities of 16 de Septiembre, Álvaro Obrégon, Bel-Ha, Conhuas, Cristobal Colón, Dos Lagunas Norte, Dos Naciones, Flores Magón, Hormiguero, Keiche Las Pailas, Ley Fomento, Mancolona, Nuevo Becal and Valentín Goméz (Campeche, Mexico) for granting us the opportunity to work in their ejidal land and for providing guide services and logistical support. We thank O. Platas-Vargas, L. Hernandez, V. Isoart, V. Corradi, K. Slater and C. Acton for their assistance in the field. We also thank JR Cedeño-Vázquez for providing insight in the early planning stages of this project and S. Mandujano-Rodríguez for provided useful suggestions on a previous version of the manuscript; A. Lopez-Cen and D. Sima-Pantí for their assistance with obtaining yearly research permits; SEMARNATCAM, J. López-Sosa, A. Balam-Koyoc, E.L. Hernández-Pérez, J.G. Hinterholzer Pino, S.E. Padilla and the team from Centro Operativo de las Direcciones de las Reservas Balam-Ku Balam-Kin for all their logistical support and assistance in fieldwork activities in Balam-Ku State Reserve; S. Henderson, J. Daw, J.C. Gutierrez, P.E. Nahuat-Cervera and the rest of the Operation Wallacea staff and student volunteers for their logistical support and assistance in the field.

### Funding

This study was possible due to the financial and logistical support of Operation Wallacea and the University of Salford through an ICase studentship to José António Lemos Barão-Nóbrega. IUCN-SSC Crocodile Specialist Group granted a Student Research Assistance Scheme (SRAS) to José António Lemos Barão-Nóbrega to assist with field equipment costs. The funders had no role in study design, data collection and analysis, decision to publish, or preparation of the manuscript.

### Grant Disclosures

The following grant information was disclosed by the authors:
Operation Wallacea and the University of Salford.
IUCN-SSC Crocodile Specialist Group.

### Competing Interests

The authors declare that they have no competing interests.

## Author Contributions

- José António Lemos Barão-Nóbrega conceived and designed the experiments, performed the experiments, analyzed the data, prepared figures and/or tables, authored or reviewed drafts of the paper, and approved the final draft.
- Mauricio González-Jaurégui conceived and designed the experiments, analyzed the data, authored or reviewed drafts of the paper, and approved the final draft.
- Robert Jehle conceived and designed the experiments, authored or reviewed drafts of the paper, and approved the final draft.

## Field Study Permissions

The following information was supplied relating to field study approvals (*i.e.*, approving body and any reference numbers):

Research permits were granted by Mexico environmental authorities "Secretaría de Medio Ambiente y Recursos Naturales" (SEMARNAT) and Comisión Nacional de Áreas Naturales Protegidas (CONANP).

## Data Availability

The raw data and code are available in the Supplemental Files.

## Supplemental Information

Supplemental information for this article can be found online at http://dx.doi.org/10.7717/peerj.12906#supplemental-information.

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
