# Peer review of "N-mixture models provide informative crocodile (Crocodylus moreletii) abundance estimates in dynamic environments"

_PeerJ, doi:10.7717/peerj.12906_

## Round 0.1 · original submission · Major Revisions

Your manuscript has now been assessed by three reviewers and myself. In general, your manuscript was well-received and your study generally considered well-done. There are a number of comments and requests from the reviewers and I have a few myself.

Reviewer 1 has a some clarifying questions that will improve the manuscript. Reviewer 1 also asks about the assumption of this species' use of wetland perimeters and I'll add that I'm curious if you should include wetland area - in addition to perimeter - in your models. Reviewer 2 also highlights a number of places where additional clarity and explanation will be helpful for your readers. Reviewer 2 also has some suggestions for additional analyses that may improve your manuscript.

Reviewer 3 has a some concerns with the amount of data used in your study and the modeling approach in general. Some of this cannot be corrected (e.g., I don't think it's feasible or reasonable to collect more data at this time) but you should be able to respond and update the manuscript to perhaps reflect additional uncertainty. I think some of the reviewer's comment may also make for additional useful and interesting discussion about ideal vs practical data and modeling in these cases and how we should interpret outcomes of various models given constraints on data while also acknowledging limitations. The reviewer offers additional analytical suggestions that are worth considering.

You write on line 249 that Calakmul represents a stronghold. But I'm curious if you would say that other estimates of the species' abundance elsewhere are very flawed and so it's unclear whether your region represents a typical abundance or a true stronghold?

This suggestion isn't mandatory but I wonder if you'd be willing to include a figure with one or multiple images of your study organism, perhaps in context of its habitat. I think doing so would be useful context for what your organism actually looks like but would also be helpful for readers to understand how detectability might vary by habitat.

In Figure 2 and it's legend, make sure to spell out completely the three different model types and show their abbreviations in parentheses to make the figure clearer.

I look forward to receiving a revised version of your manuscript.

·

Basic reporting

Overall, the paper is well-written and clear, the results are placed in the appropriate context, the background information is thorough, and the figures and raw data pass the sniff test.

Experimental design

I find the research question to be very interesting because many crocodilian species are threatened and it’s difficult to develop appropriate management plans because getting accurate population estimates can be difficult. This paper lays out a clear road map for how to generate reliable population abundance estimates for crocodilians. The experimental design is mostly solid, but there are a couple of points that need clarification (see below). In general, a bit more detail in the survey methods would be very helpful.

Validity of the findings

The findings are valid. I also have a few minor comments that hopefully will be useful (see below).

I enjoyed reading this manuscript and I hope to see it published.

Additional comments

Line 143: How did the authors estimate total length from eyeshine? Need to explain their methods in a bit more detail here.

Lines 172-173: What statistical test are the authors reporting the results for here?

Lines 176-177: How were hydric state and vegetation cover determined?

Lines 177-179: Why was this necessary to do?

Line 189: Should be “we” instead of “we’ve”

Lines 196-198 and Lines 139-140: Seems like there’s a big assumption in this study: that the crocodiles are only found on the shoreline. Is that really true for this species? Did the authors survey open water at all? If so, please describe. Also, how big are each of the surveyed aguadas in terms of surface area and perimeter? Would be good to know the size range.

Lines 196-198: Why not just add up all the local abundance estimates for each water body? Why do the authors have to do this extrapolation at all?

Lines 327-331: The use of the terms “genetically pure” and “genetic pollution” here is a bit odd to me. It reads like these are value judgements of different crocodile populations based on their potential for hybridization. Please consider rephrasing.

Lines 377-379: What is this reference? Is this a document published by the website ResearchGate? If so, is it actually citable?

Reviewer 2 ·

Basic reporting

The manuscript is well written, well-documented, and sufficient field background/context was provided. I have some minor recommendations to improve the clarity of the information provided in the methods section, that will help readers to follow the rationale of the manuscript. Authors must include clue information such as c-hat of every important model so readers can judge the efficiency of models. I suggest sending figure 5 to supplementary material as I don't see the relevance of it. I also suggest running some extra models to shed some clarity about the effect outliers can bring to the results presented. Authors can find detailed comments in the document attached.

Experimental design

No comment

Validity of the findings

No comment

Additional comments

No comment

Annotated reviews are not available for download in order to protect the identity of reviewers who chose to remain anonymous.

Reviewer 3 ·

Basic reporting

No captions for figures. Figures need more explanation to fully evaluate.

Several minor grammatical or word choice issues. See attached document.

Lack of consistency in using the SI abbreviation m for meters/meteres.

Experimental design

In the methods it is not clear how often each site was surveyed, but I was able to determine from the data provided in the supplement that each site was surveyed 2-7 times. What concerns me the most about this, is the fact that this takes place over several years and during different seasons within a year. One important assumption of the N-mixture model you are using is that the population at each site is closed to changes in abundance. Thus all differences in counts over time are due to changes in detection probability. However, here you are using counts spanning up to 4 years and do not have any repeat surveys within a season. This is potentially an issue and should at least be explained in the methods so that readers can understand that the assumptions of the model may be being violated.
A better way to handle the question of changes in abundance over time would be to use the dynamic version of this model (pcountOpen in unmarked). However, you would still have the issue of no repeat samples within a season.
I think it is great to compare the hierarchical model with the MPS and VFM methods, but I would like to see more detail on these. For instance, when you have 7 counts made over 4 years for a site, what value is used for the MPS? Is it the highest value during that period or the mean of the 7 values? If it is the sum across the 7 samples at a site, doesn’t that risk counting a single individual more than once? I also do not understand exactly what relative abundance is as you use it and would like to see that better defined.
When it comes to the hierarchical modeling, besides my concerns over the use of data from a long time span in a closed model, I also have concerns about the way models were compared. It is worthwhile to compare the model types (i.e. parameter distributions) but it is not appropriate to compare between them using AIC. AIC should only be used within a model type to rank models not across model types. This is because the model likelihoods are different, and there is no theoretical basis for comparing models with different likelihoods. This means that Table 1 should be re-done and any reference to comparing the models using AIC should be corrected.
In addition to providing the total estimates for each method in Table 2, I would like to see the estimates of abundance for each waterbody using the various methods. To me it would be much easier to understand the differences between the models that way. Maybe this could go in a supplement?

Validity of the findings

The findings may be valid, but the model selection method needs to be revised. Also, the fact that the models used are unlikely to meet the assumptions of the model chosen needs to be addressed. At a minimum the authors should discuss this fact. I would consider throwing out the negative binomial model all together.

We must see more details on the methods for the Generalized Linear Mixed Model and the error estimation for the estimates.

Additional comments

I found this to be an interesting manuscript with some valuable information. However, I do see some shortcomings that need to be addressed. I believe that the hierarchical modeling used in this paper is a sound approach, but the dataset used in the analysis are very sparse for to put such high confidence in these estimates. It seems that the main point of this paper is to support the use of the N-mixture model for this system, but I remain unconvinced that it is completely appropriate.
On the manuscript I have made numerous comments in specific places that should be addressed. Here I will try to relay my general concerns.

Annotated reviews are not available for download in order to protect the identity of reviewers who chose to remain anonymous.

---

## Round 0.2 · accepted · Accept

Thank you for your thorough revision of this manuscript in response to the reviewers.